# First-Principles Study of Electronic Structure and Optical Properties of Ni-Doped Bi$_4$O$_5$Br$_2$

Hong Sheng [1,†], Xin Zhang [2,†], Shiheng Xin [2], Hui Shi [2], Gaihui Liu [2], Qiao Wu [3], Suqin Xue [3], Xiaoyang Wang [2], Tingting Shao [2], Yang Liu [2,*], Fuchun Zhang [2,*] and Xinghui Liu [4,5,*]

1. College of Mathematics & Physics, Weinan Normal University, Weinan 714000, China; wnshenghong@163.com
2. School of Physics and Electronic Information, Yan'an University, Yan'an 716000, China; zx1554485388@163.com (X.Z.); shihengxin369@163.com (S.X.); shihui@yau.edu.cn (H.S.); liugaihui@yau.edu.cn (G.L.); yettawang15@163.com (X.W.); shaotingting@yau.edu.cn (T.S.)
3. Network Information Center, Yan'an University, Yan'an 716000, China; wq@yau.edu.cn (Q.W.); ydxsq@yau.edu.cn (S.X.)
4. Science and Technology on Aerospace Chemical Power Laboratory, Xiangyang 441003, China
5. Hubei Institute of Aerospace Chemotechnology, Xiangyang 441003, China
* Correspondence: yangliu06152022@163.com (Y.L.); yadxzfc@yau.edu.cn (F.Z.); liuxinghui119@gmail.com (X.L.)
† These authors contributed equally to this work.

**Abstract:** In this study, we comprehensively explored the electronic structure and optical properties of Ni-doped Bi$_4$O$_5$Br$_2$ through first-principles computational calculations. By calculating its electronic structure and band characteristics, we investigated the impact of Ni doping on the photocatalytic performance of Bi$_4$O$_5$Br$_2$. The computational results indicated that Ni doping significantly altered the band structure of Bi$_4$O$_5$Br$_2$, leading to a reduction in the band gap width. The band gap for undoped Bi$_4$O$_5$Br$_2$ was 2.151 eV, whereas the Ni-doped system exhibited a smaller band gap, directly indicating its enhanced visible light absorption capacity and facilitating the effective separation of photo-generated electron–hole pairs. Through analysis of 2D charge density maps, we observed changes in chemical bonding induced by Ni doping. The shortening of Ni-O bonds suggested increased bond strength, consistent with the observed reduction in cell volume. These findings provide a theoretical foundation for understanding the mechanisms behind the enhanced photocatalytic hydrogen production performance in Ni-doped Bi$_4$O$_5$Br$_2$, offering valuable insights for the design and optimization of highly efficient photocatalytic materials.

**Keywords:** Ni-doped; Bi$_4$O$_5$Br$_2$; first-principles calculations; photocatalysis

## 1. Introduction

In 1976, Japanese scientist Fujishima made a groundbreaking discovery that TiO$_2$ could catalyze solar energy conversion through photocatalysis [1]. This discovery sparked widespread interest among scientists, leading to significant advancements in energy and environmental management [2,3]. Photocatalysis has demonstrated immense potential in various applications, including water splitting to produce hydrogen [4], CO$_2$ reduction into organic compounds like methane, methanol, and formic acid [5], and ammonia synthesis from N$_2$ [6], providing a clean and sustainable alternative to conventional methods.

Furthermore, photocatalysis plays a crucial role in eco-friendly water pollution control, preventing secondary pollution [7]. Various materials have been explored as photocatalysts, including common sulfide photocatalysts like CdS [8], semiconductor oxide materials such as TiO$_2$ and ZnO [9,10], as well as materials like g-C$_3$N$_4$ and bismuth oxide semiconductors (Bi$_x$O$_y$H$_z$, where H=Cl, Br, I) [11,12]. Additionally, research has extended to encompass metal-organic framework materials and covalent organic framework materials [13–15]. These materials have encountered limitations related to band gaps, light absorption ranges,



and electron–hole recombination during photocatalysis [16,17]. Researchers have employed diverse strategies to address these challenges, including constructing heterojunctions, loading materials to inhibit electron–hole recombination, and doping to modify the energy band structure [18–22]. Among these approaches, doping stands out as the simplest and most effective method [23–25].

Bismuth oxybromide ($Bi_4O_5Br_2$), as a ternary (V-VI-VII) semiconductor with the molecular formula BiOX (X = Cl, Br, I), has garnered significant attention in the field of photocatalysis due to its suitable band gap, unique two-dimensional (2D) layered structure, and broad spectrum light responsiveness [26–31]. Among these BiOX materials, BiOBr, with its appropriate band structure, has been a focal point of interest, yet there is still room for improvement in its catalytic activity and light absorption range [32–36]. In particular, $Bi_4O_5Br_2$ nanostructures have drawn our attention due to their broader visible light absorption edge and negative conduction band position when compared to pure BiOBr materials [37–39]. However, for the practical application of $Bi_4O_5Br_2$ in engineering, there are still some limitations to address. For instance, it exhibits a relatively high charge carrier recombination rate and a larger band gap, which restricts its photocatalytic efficiency and also limits its absorption of visible light [40].

Despite the abundance of research on bismuth oxybromide ($Bi_4O_5Br_2$) photocatalytic materials and doping modifications, there has been relatively limited systematic theoretical investigation into their electronic structure and optical properties [41–44]. Recently, in the work by Wang et al., the enhancement of photocatalytic performance has been achieved by doping Ni into BiOBr to increase the unit cell dipole moment and induce spin polarization, thereby augmenting the built-in electric field [45]. Therefore, this study employs density functional theory-based pseudopotential plane-wave methods to systematically calculate the electronic structure, density of states, complex dielectric function, absorption spectra, and other properties of Ni-doped $Bi_4O_5Br_2$. The aim is to theoretically investigate how Ni doping enhances the photocatalytic performance of $Bi_4O_5Br_2$. This research provides crucial theoretical insights for the design and application of similar studies in the future.

## 2. Computational Methods

$Bi_4O_5Br_2$ belongs to the $P2_1$ space group with $a$ = 10.94 Å, $b$ = 5.65 Å, $c$ = 14.60 Å, $\alpha = \gamma = 90°$, $\beta = 98.02°$ [46]. To calculate exchange-correlation energy, we applied the Perdew-Burke-Ernzerh (PBE) generalized gradient approximation (GGA) based on density generalization theory using the CASTEP code [47–49]. Geometric optimization employs the Broyden–Fletcher–Goldfarb–Shanno (BFGS) method, and core electron interactions are computed with OTFG ultrasoft pseudopotential [50,51]. For geometric optimization and property calculations, we used a 6 × 6 × 2 k-point mesh and an energy cut-off of 580 eV. During atomic relaxation, energy convergence tolerance was set to no more than $1 \times 10^{-5}$ eV/atom, and atomic forces were limited to less than 0.03 eV/Å, with maximum stress and displacement capped at 0.05 GPa and 0.001 Å.

All computational models were constructed using $Bi_4O_5Br_2$ (Figure 1a) original cells, which contained 16 Bi atoms, 20 O atoms, and 8 Br atoms. The model of the single-doped system is to replace one Bi atom with Ni. In our calculations, we only considered nickel replacement doping $Bi_4O_5Br_2$ (Figure 1b) because the radius and valence states of nickel atoms favor replacement doping rather than gap doping [52].

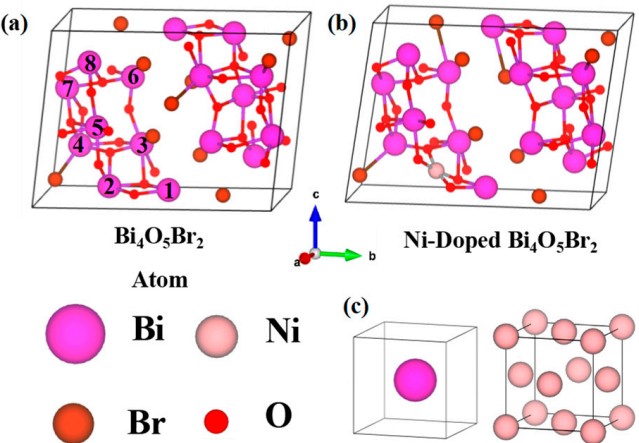

**Figure 1.** (**a**) $Bi_4O_5Br_2$ (**b**) Ni-Doped $Bi_4O_5Br_2$ (**c**) The structure used to calculate the chemical potential of Bi and Ni.

## 3. Results and Discussion

### 3.1. Structure Optimization

After Ni replaces one Bi atom in the $Bi_4O_5Br_2$ primary cell, the lattice constants obtained were *a* = 5.74, *b* = 10.99, and *c* = 14.73 nm, which are 0.05 nm smaller than those obtained by calculating $Bi_4O_5Br_2$ for a, 0.15 nm smaller than for b, and 0.27 nm larger than for c.

All the parameters of the two structures are listed in Table 1, and the main reason for the analysis is that the radius of the Ni atoms is smaller than that of the Bi atoms and that Ni is less electronegative, forming stronger covalent bonds with O after replacing Bi atoms. This is mainly due to the fact that the radius of Ni atom is smaller than that of Bi, and Ni is less electronegative and forms stronger covalent bonds with O after the replacement of Bi atoms. The calculated Ni-O bond length is shorter than the Bi-O bond, which increases the lattice constant. Chen et al. predicted through DFT calculations that by adjusting the reaction solution pH during the solvent–thermal synthesis process, the crystal lattice parameter b of $Bi_2MoO_6$ could be shortened, inducing a larger internal polarization [53]. Similarly, in our study, we achieved a reduction in lattice parameters through Ni doping, which significantly affects the distribution of electron density, thereby inducing the distribution of electron–hole pairs and indirectly improving the photocatalytic performance.

**Table 1.** Lattice parameters before and after optimization.

|  | a | b | c | α | β | γ | Volume |
|---|---|---|---|---|---|---|---|
| $Bi_4O_5Br_2$ | 5.79 | 11.14 | 15.00 | 82.43 | 89.99 | 90.01 | 960.41 |
| Ni-$Bi_4O_5Br_2$ | 5.74 | 10.99 | 14.73 | 82.35 | 91.05 | 87.87 | 950.17 |

To assess the comparative stability of the analyzed systems, we computed their doped formation energies using the following formula:

$$E_f = E_{(Ni-Bi_4O_5Br_2)} - E_{(\text{pure}-Bi_4O_5Br_2)} + n_{Bi}\mu_{Bi} - n_{Ni}\mu_{Ni} \tag{1}$$

where;

- $E_{(Ni-Bi_4O_5Br_2)}$ is the total energy of the doped $Bi_4O_5Br_2$,
- $E_{(\text{pure}-Bi_4O_5Br_2)}$ is the total energy of $Bi_4O_5Br_2$ without doping,
- $n_{Bi}$ and $n_{Ni}$, represent the quantities of added or removed atoms of Bi and Ni, respectively, in the assembly of various cells,

- and $\mu_{Bi}$ and $\mu_{Ni}$ are the chemical potentials of, respectively; determined through DFT calculations as illustrated in Figure 1c. Each of $\mu_{Bi}$ and $\mu_{Ni}$ corresponds to the energy per atom in their respective bulk crystals, $\mu_{Bi}$ and $\mu_{Ni}$.

Simultaneously, we computed doping formation energies for Ni separately (refer to Table 2). The lower the doping formation energy, the greater the stability of the resulting structure. Therefore, the structure with the eighth site is deemed optimal.

**Table 2.** Calculated doping formation energy at different locations.

| Site | 1 | 2 | 3 | 4 | 5 | 6 | 7 | 8 |
|---|---|---|---|---|---|---|---|---|
| Formation energy/eV | 1.76 | 1.70 | 2.42 | 2.44 | 2.62 | 2.34 | 2.71 | 1.91 |

### 3.2. Electronic Structures

The depicted energy band structures in Figure 2 illustrate the characteristics of both undoped and Ni-doped materials, with the Fermi energy reference point set at 0 eV. It is important to acknowledge that, in the realm of density functional theory (DFT), the GGA-PBE method is renowned for its limitations, particularly in its tendency to underestimate the material's band gap when compared to experimental observations. Nevertheless, despite this minor drawback, the obtained band gap remains adequate for conducting a comprehensive exploration of electronic states and investigating optical properties in subsequent analyses.

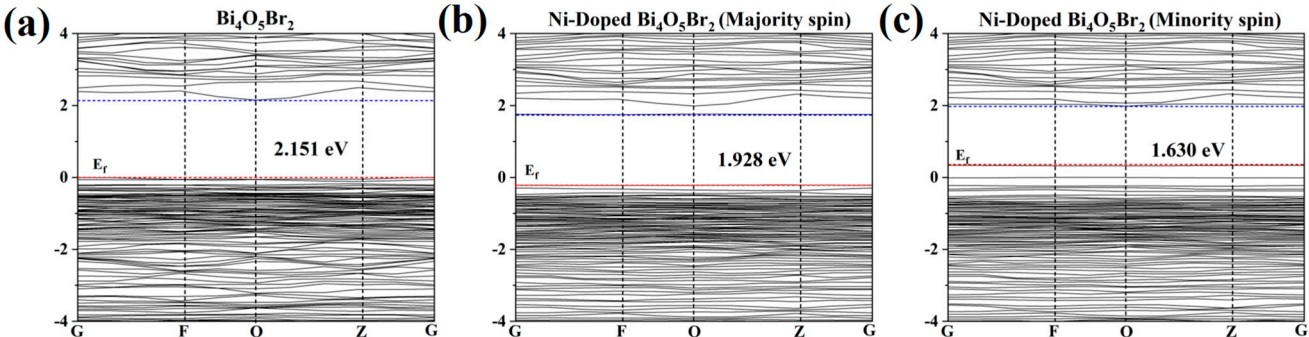

**Figure 2.** Energy band (**a**) $Bi_4O_5Br_2$, after considering spin (**b**) majority spin (**c**) minority spin.

Furthermore, it is worth noting that the computed band gap for undoped $Bi_4O_5Br_2$, as depicted in Figure 2a, stands at 2.151 eV, a mere 0.17 eV deviation from the experimentally measured band gap width. While the predictions of PBE are not entirely accurate, it enables a faster prediction of energy trends following structural changes without impacting the analysis of band structures and electronic configurations [54]. As evident in Figure 2, there is a discernible decline in band gap energy as we transition from the undoped system to the Ni-doped system. This trend indirectly implies a gradual enhancement of the material's light absorption capability. In a recent work, Mao et al., using UV–vis diffuse reflectance spectra and density function theory (DFT) calculations, indicated that $Bi_4O_5Br_2$ possessed stronger visible light adsorption, which is beneficial to effectively activate molecular oxygen to produce [55]. This implies that the incorporation of Ni can significantly enhance the light absorption capability of $Bi_4O_5Br_2$, promoting the transfer of electrons and holes as well as the generation of superoxide radicals. Combining these improvements in the catalyst preparation process, the performance of the photocatalyst can be greatly elevated.

To delve deeper into the examination of doping's impact on electronic properties, we carried out calculations for the partial density of states (PDOS) and electron density distributions around the Fermi energy level for both undoped and doped $Bi_4O_5Br_2$, as illustrated in Figure 3. Regarding undoped $Bi_4O_5Br_2$, the partial density of states (PDOS) is presented in Figure 3a. In this context, it is noteworthy that the valence band maxima

(VBM) primarily originate from the 2p orbitals of oxygen (O) and the 4p orbitals of bromine (Br), while the conduction band minima (CBM) are predominantly influenced by the 2p orbitals of oxygen (O) and the 6p orbitals of bismuth (Bi).

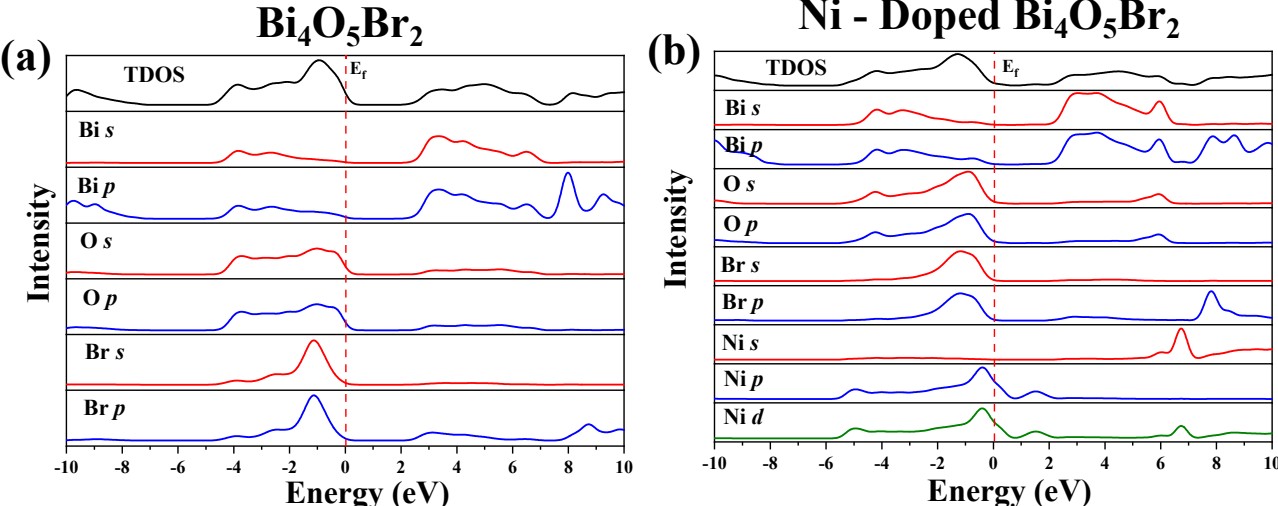

**Figure 3.** Total state density (TDOS) and projected state density (PDOS) of (**a**) $Bi_4O_5Br_2$ and (**b**) Ni-doped $Bi_4O_5Br_2$.

In the case of Ni-doped $Bi_4O_5Br_2$, as depicted in Figure 3b, the valence band maximum (VBM) primarily arises from the O-2p orbitals with contributions from Ni-3d orbitals. Conversely, the conduction band minimum (CBM) is predominantly governed by the Bi-6p orbitals. Notably, the introduction of Ni doping, with Ni-3d peaks occurring within the energy range of 0–2 eV, imparts a semi-metallic character to $Bi_4O_5Br_2$. After Ni doping, a sharp peak in the vicinity of 6 eV is observed in the O s and p orbitals. This is likely due to the formation of a stable chemical bond between Ni and O, resulting in a significant electron density near this energy. This sharp peak may have a crucial impact on electron transport and photocatalytic performance of the material. Additionally, an enhanced density of states (DOS) is observed in the range of 7–10 eV for Bi p orbitals and Br p orbitals, attributed to the contribution of Ni d orbitals. Interaction between Ni d orbitals and Bi/Br likely induces an increase in the electron density of these orbitals. This interaction may lead to the formation of new electronic states, influencing the electronic structure of the catalyst. Consequently, this alteration reduces the energy gap between the valence band and conduction band, resulting in a redshift of the optical absorption edge. This, in turn, enhances the photocatalytic efficiency of the doped system.

In Figure 4a–c,f–h, we present the 2D charge density maps for $Bi_4O_5Br_2$ and Ni-doped $Bi_4O_5Br_2$, respectively. Upon analyzing various cross-sections, it becomes evident that the introduction of Ni in place of Bi (Figure 4d,e) results in a reduction in bond length, indicating that the Ni-O bond is stronger compared to the Bi-O bond. This observation aligns with the findings in Table 1, which indicate a decrease in cell volume following Ni atom doping. As illustrated in Figure 4, the differences in bonding properties between the atoms in the undoped and doped cases are relatively subtle. Furthermore, as demonstrated in Figure 4i, we explored the impact of Ni doping on the electron transfer properties of the $Bi_4O_5Br_2$ samples by utilizing the charge density difference calculated through DFT. In this representation, the yellow color signifies a gain of electrons, while the blue color indicates electron loss.

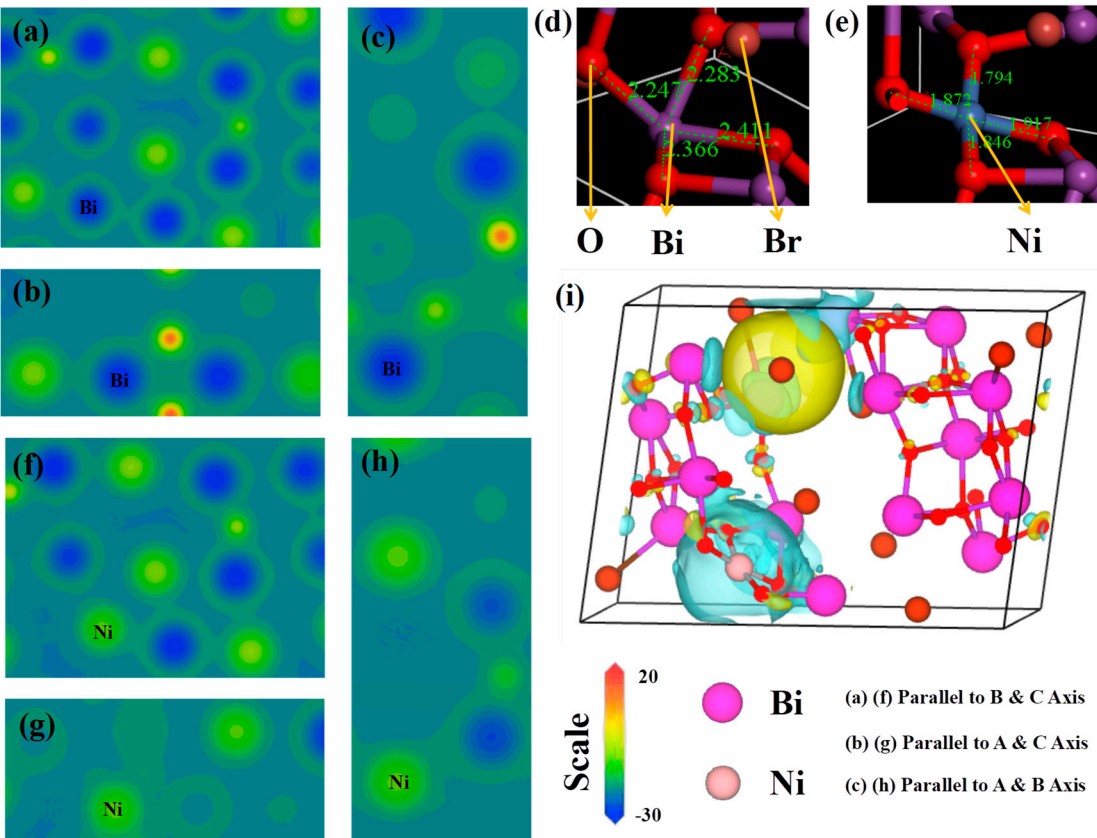

**Figure 4.** 2D charge-density plots (**a**–**c**) without Ni and (**f**–**h**) with Ni doped sections in different directions. (**d**,**e**) The change in bond length of Ni-O and Bi-O after replacement doping. (**i**) 3D charge density map after doping.

It is noteworthy that the difference in charge density highlights an electron-rich environment around the surface of Br atoms. Observing Figure 4i allows for us to conclude that, during the photocatalytic process, the built-in electric field formed after Ni doping suppresses the recombination of electrons (e$^-$) and holes (h$^+$), thereby enhancing charge separation efficiency. This enables sustained involvement of photo-generated electrons in reduction reactions, while photo-generated holes participate in oxidation reactions, ultimately improving the photocatalytic performance.

### 3.3. Optical Properties

Ni doping introduces impurity energy levels that markedly enhance light absorption. The simulated absorption spectra of Bi$_4$O$_5$Br$_2$ and Ni-doped Bi$_4$O$_5$Br$_2$ are depicted in Figure 5a. The imaginary part of $\varepsilon_2(\omega)$ related to the optical absorption of the photocatalytic materials is shown in Figure 5b. The absorption coefficient is expressed as follows:

$$\alpha(\omega) = (\sqrt{2})\omega \left[\sqrt{\varepsilon_1(\omega)^2 + \varepsilon_2(\omega)^2} - \varepsilon_1(\omega)\right]^{\frac{1}{2}} \tag{2}$$

where $\varepsilon_1(\omega)$ and $\varepsilon_2(\omega)$ are the real and imaginary parts of the dielectric function, respectively, and both depend on the optical frequency ($\omega$). When the dielectric function $\varepsilon_2$ is not zero, it can be determined that the absorption coefficient is positively correlated with it. Doping induces a redshift in absorption peaks, expanding the absorption range. In the low-energy region, Ni doping significantly enhances absorption, favorably influencing photocatalytic performance.

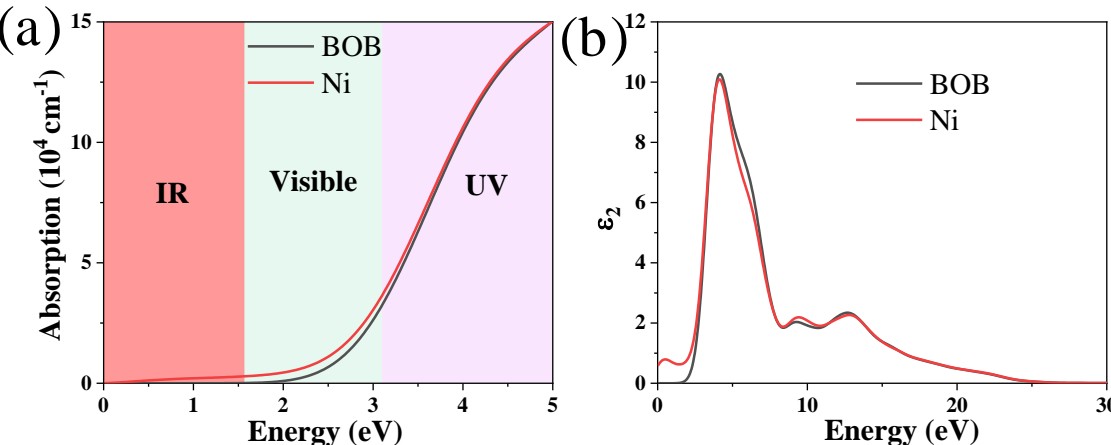

**Figure 5.** Imaginary part of (**a**) light absorption range (**b**) dielectric function of $Bi_4O_5Br_2$ and Ni-$Bi_4O_5Br_2$.

The light absorption of $Bi_4O_5Br_2$ in the visible range (1.6–3.1 eV) was poor. Ni doping resulted in a significant enhancement of the light absorption of $Bi_4O_5Br_2$ in the visible range and extended the absorption range from the visible region to the infrared region. It is proved that Ni doping is favorable for the photocatalytic activity enhancement of $Bi_4O_5Br_2$.

### 3.4. Photocatalytic Properties

In a photocatalytic reaction, semiconductor materials initially absorb the energy of photons, which must exceed or match the material's band gap energy level ($E_g$). This photon-induced excitation results in the generation of photoexcited electron–hole pairs within the semiconductor. Photoexcited electrons transition to the conduction band, attaining higher energy levels, while photoexcited holes remain in the valence band. Subsequently, these photoexcited charge carriers (electrons and holes) propagate through the semiconductor material, either driven by an applied electric field or moving through diffusion toward the material's surface. This step is crucial as it facilitates the transport of photoexcited carriers to the catalytic reaction sites. However, during their migration, photoexcited electrons and holes may occasionally recombine, releasing previously absorbed energy, typically in the form of heat or light. Upon reaching the surface of the semiconductor and the catalytic sites, photoexcited electrons and holes participate in the catalytic reaction. Typically, photoexcited electrons are involved in reduction reactions, while photoexcited holes take part in oxidation reactions. This surface-driven redox chemistry ultimately initiates and sustains the catalytic reactions. These catalytic reactions encompass a broad range of processes, including gas conversion, water decomposition, organic compound degradation, and more. In these reactions, chemical energy is usually released, leading to the production of desired products.

Hence, we have derived the CBM and VBM of $Bi_4O_5Br_2$ through a synergistic approach that combines theoretical insights with DFT calculations. This comprehensive approach enhances our ability to make more accurate predictions regarding $Bi_4O_5Br_2$ photocatalytic performance. The CBM and VBM potentials of undoped $Bi_4O_5Br_2$ are empirically calculated using the following formulas [56]:

$$E_{CBM} = -\frac{1}{2}E_g + \chi_{Bi_4O_5Br_2} + E_0 \tag{3}$$

$$E_{VBM} = +\frac{1}{2}E_g + \chi_{Bi_4O_5Br_2} + E_0 \tag{4}$$

$$\chi_{Bi_4O_5Br_2} = (\chi^4_{Bi}\chi^5_O\chi^2_{Br})^{\frac{1}{11}} \tag{5}$$

where $E_g$ is the band gap energy, $\chi Bi_4O_5Br_2$ is the absolute electronegativity of $Bi_4O_5Br_2$. $\chi Bi$, $\chi Br$ and $\chi O$ are the Pearson absolute electronegativity of, respectively. According to the literature: $\chi Bi = 4.69$, $\chi Br = 7.59$ and $\chi O = 7.54$ [57]. The parameter $E_0$ serves as the scale factor that connects the reference redox level to the absolute vacuum scale [56]. The obtained CBM of $Bi_4O_5Br_2$ is 0.77 and VBM is 2.92. While for the $Bi_4O_5Br_2$ after Ni doping the CBM and VBM are 0.89 and 2.81, respectively. In the case of Ni-doped $Bi_4O_5Br_2$, the CBM is shifted downward by 0.12 eV with respect to $Bi_4O_5Br_2$, thus greatly improving the photoreduction capacity. Furthermore, the VBM values of the doped system span the oxidation-reduction potential of hydroxyl radicals. This is consistent with the analysis in the electronic structure section, where changes in the valence band and conduction band are evident in the generation of free radical groups. This can significantly improve the catalytic performance of the catalyst.

Therefore, the mechanism underlying the enhanced photocatalytic activity can be elucidated as follows. Firstly, Ni introduces impurity energy levels with both upper and lower spin in the bandgap of $Bi_4O_5Br_2$ after doping, altering the electron transition pathways and significantly reducing the bandgap of $Bi_4O_5Br_2$. Secondly, the optical absorption performance of the Ni-$Bi_4O_5Br_2$ composite material not only exhibits a pronounced redshift but also shows enhanced absorption intensity. Thus, this catalysis even holds potential for infrared reactions. In comparison to pristine $Bi_4O_5Br_2$, these two advantages can significantly increase the excitation rate of photo-generated electron–hole pairs. The effective spatial separation of oxidation and reduction centers markedly suppresses the recombination of electron–hole pairs, thereby improving charge carrier utilization. The combined effect of these factors endows the Ni-$Bi_4O_5Br_2$ composite material with outstanding photocatalytic performance.

## 4. Conclusions

In order to explore the physical origin of the experimentally observed doping enhancement on the photocatalytic performance of $Bi_4O_5Br_2$, we conducted a comprehensive study of the electronic structure and optical properties of Ni-doped $Bi_4O_5Br_2$ through first-principles computational calculations. The computational results indicate that Ni doping significantly alters the band structure of $Bi_4O_5Br_2$. The band gap for undoped $Bi_4O_5Br_2$ is 2.151 eV, whereas the Ni-doped system exhibits a smaller band gap, directly indicating its enhanced visible light absorption capacity. In comparison to the undoped system, Ni doping leads to more efficient separation of electrons and holes, further promoting the photocatalytic performance. Through the analysis of 2D charge density maps, we observed changes in chemical bonding induced by Ni doping. The shortening of Ni-O bonds suggests increased bond strength, consistent with the observed reduction in cell volume.

Furthermore, our analysis using DFT-computed charge density difference maps revealed that Ni doping alters the electron transfer and changes the electron-rich environment around Br atoms on the surface. We can conclude that, during the photocatalytic process, the built-in electric field formed after Ni doping suppresses the recombination of electrons ($e^-$) and holes ($h^+$), thereby enhancing charge separation efficiency. In terms of light absorption, Ni doping significantly enhances the absorption of $Bi_4O_5Br_2$ in the visible range, extending the absorption range into the infrared region. This enhancement is not only theoretically supported but also quantitatively contributes to the improved photocatalytic activity of $Bi_4O_5Br_2$. This study provides concrete guidance for the design of efficient photocatalytic materials and underscores the effectiveness of Ni doping as a strategy to enhance photocatalytic performance.

**Author Contributions:** F.Z. and X.L. supervised the research; H.S. (Hong Sheng) and Y.L. designed the experimental program; H.S. (Hong Sheng) and Y.L. completed the experimental preparation and wrote the first draft; F.Z. provided funding acquisition and revised data and papers; X.Z., S.X. (Shiheng Xin), G.L., Q.W., S.X. (Suqin Xue), X.W., T.S. and H.S. (Hui Shi) et al. analyzed and confirmed the data; All authors participated in analysis of the experimental data and discussions of the results, as well as editing the manuscript. All authors have read and agreed to the published version of the manuscript.

**Funding:** This research was funded by the National Natural Science Foundation of China (No: 62264015), 2023 Innovation and Entrepreneurship for Undergraduates (2023107 19040).

**Data Availability Statement:** Data are contained within the article.

**Conflicts of Interest:** There are no conflicts to declare.

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
