# Peer review of "First-Principles Study of Electronic Structure and Optical Properties of Ni-Doped Bi4O5Br2"

_coatings, doi:10.3390/coatings14010067_

Round 1
Reviewer 1 Report
Comments and Suggestions for Authors
Manuscript Number: coatings-2796843
Full Title: First principles study of electronic structure and optical properties of Ni-doped Bi4O5Br2.
Type: Article.
Although the manuscript quality has immature features, this work actually provides lots of effective data. Publication of these data in public journal media would have good contributions to the related research fields. From the latter positive viewpoints, I may recommend publication of this work in coatings. However, several revisions are necessary.
1- In the section of Introduction, the authors did not mention why doping Ni was introduced into Bi4O5Br2 samples.
2- The image is not clear (Fig. 2). It will also create issue to believe the originality.
3- In order to show superiority of the current materials, discussions on comparisons of performance over the other related materials reported in the past literatures are necessary. Please add descriptions on such comparisons.
4- What is the relationship between microstructure properties and photocatalytic properties of the studied samples?
5- Some quantitative information required in the abstract and conclusion.
6- The Authors should also proofread their manuscript (some spelling and grammar errors).
Comments on the Quality of English LanguageThe Authors should also proofread their manuscript (some spelling and grammar errors).
Reviewer 2 Report
Comments and Suggestions for Authors
The manuscript "First principles study of electronic structure and optical properties of Ni-doped Bi4O5Br2" is quite interesting. However, this manuscript needs minor improvement to be published in the Coatings journal. Here are some major improvements that need to be considered:
- The manuscript shows around 22% plagiarism. The percentage is acceptable. Kindly see the similarity report.
- It is better to change the title to be (First-principles study of electronic structure and optical properties of Ni-doped Bi4O5Br2).
- The abstract and conclusion parts should be modified to give a clear idea of the results.
- The author should represent and highlight the novelty of the work in the manuscript.
- Related to Fig.3, kindly improve the resolution to be clear for the readers.
- The authors need to correlate the structure of the materials and the photocatalytic performance.
- Some typos and errors need to be corrected within the manuscript.

Comments on the Quality of English LanguageMinor editing of English language required.
Reviewer 3 Report
Comments and Suggestions for Authors
Here are some basic queries and suggestions to consider when submitting a review for the paper entitled "First principles study of electronic structure and optical proper-2 ties of Ni-doped Bi4O5Br2”: I would like to recommend it for publication after minor revision. The following issues need to be considered:
1. In the introduction section, Authors should highlight some recent works reported in the literature related to this work. It would be beneficial for the introduction to briefly mention the potential practical applications or implications of improving the photocatalytic performance of Bi4O5Br2 through Ni doping, connecting the theoretical findings to real-world scenarios.
22. "In Equation 1, it seems there are formatting issues with superscripts. Could the authors kindly provide clarification and rectify these elements throughout the entire manuscript to enhance overall clarity and accuracy?"
3. On page 4, Regarding the computed band gap for undoped Bi4O5Br2, can the authors elaborate on the significance of the 0.17 eV deviation from the experimentally measured band gap width? Are there any implications for the material's properties or applications?
4. In the discussion of the partial density of states (PDOS) and electron density distributions around the Fermi energy level, could the authors provide insights into how these electronic properties are influenced by doping in both undoped and doped Bi4O5Br2? Are there any observed changes in the electronic properties, such as shifts in the PDOS or electron density distributions, that are specifically attributable to Ni doping in Bi4O5Br2?
5. Could the authors elaborate on the specific mechanisms involved in the photocatalytic reaction process, such as carrier generation, migration, potential recombination, and surface-catalyzed redox reactions, to provide a more detailed understanding of the overall process?
6. In the comparison of CBM and VBM values for undoped and Ni-doped Bi4O5Br2, what are the implications of the downward shift in CBM by 0.12 eV after Ni doping? How does this improvement in photoreduction capacity contribute to the overall photocatalytic performance?
7. Several typographical errors and sentences with ambiguous meanings were found in the manuscript. Please read the article and make necessary corrections in its entirety.

Comments on the Quality of English LanguageMinor editing of English language required.
Round 2
Reviewer 1 Report
Comments and Suggestions for Authors
Manuscript Number: coatings-2796843-R1
Full Title: First principles study of electronic structure and optical properties of Ni-doped Bi4O5Br2.
Type: Article.
After authors' careful revision, the manuscript seems fluent and readable now. Most of the grammar mistakes were amended. The contents, discussions and the conclusions have been revised in this version. May be accepted for publication.